# Impact of Remote Ischemic Preconditioning Conducted in Living Kidney Donors on Renal Function in Donors and Recipients Following Living Donor Kidney Transplantation: A Randomized Clinical Trial

**DOI:** 10.3390/jcm8050713

**Published:** 2019-05-20

**Authors:** Ji-Yeon Bang, Sae-Gyeol Kim, Jimi Oh, Seon-Ok Kim, Yon-Ji Go, Gyu-Sam Hwang, Jun-Gol Song

**Affiliations:** 1Department of Anesthesiology and Pain Medicine, Asan Medical Center, University of Ulsan College of Medicine, 88, Olympic-ro 43-gil, Songpa-gu, Seoul 05505, Korea; jyounbang@gmail.com (J.-Y.B.); saegyul@naver.com (S.-G.K.); ohzie82@gmail.com (J.O.); goyunji426@gmail.com (Y.-J.G.); kshwang@amc.seoul.kr (G.-S.H.); 2Department of Clinical Epidemiology & Biostatistics, Asan Medical Center, University of Ulsan College of Medicine, 88, Olympic-ro 43-gil, Songpa-gu, Seoul 05505, Korea; seonok@amc.seoul.kr

**Keywords:** living donor kidney transplantation, remote ischemic preconditioning, residual renal function, chronic kidney disease

## Abstract

Although remote ischemic preconditioning (RIPC) has been shown to have renoprotective effects, few studies have assessed the effects of RIPC on renal function in living kidney donors. This study investigated whether RIPC performed in living kidney donors could improve residual renal function in donors and outcomes in recipients following kidney transplantation. The donors were randomized into a control group (*n* = 85) and a RIPC group (*n* = 85). The recipients were included according to the matched donors. Serum creatinine (sCr) concentrations and estimated glomerular filtration rate (eGFR) were compared between control and RIPC groups in donors and recipients. Delayed graft function, acute rejection, and graft failure within one year after transplantation were evaluated in recipients. sCr was significantly increased in the control group (mean, 1.13; 95% confidence interval (CI), 1.07–1.18) than the RIPC group (1.01; 95% CI, 0.95–1.07) (*p* = 0.003) at discharge. Donors with serum creatinine >1.4 mg/dL at discharge had higher prevalence of chronic kidney disease (*n* = 6, 26.1%) than donors with a normal serum creatinine level (*n* = 8, 5.4%) (*p* = 0.003) after one year. sCr concentrations and eGFR were similar in the RIPC and control groups of recipients over the one-year follow-up period. Among recipients, no outcome variables differed significantly in the RIPC and control groups. RIPC was effective in improving early renal function in kidney donors but did not improve renal function in recipients.

## 1. Introduction

Kidney transplantation (KT) is the most effective treatment for many patients with kidney failure [1], with living donor KT having a higher survival rate than deceased donor KT [2]. Kidney donors, however, are at risk of reduced residual renal function [3,4,5], as the abrupt loss of renal mass and resultant adaptive hyperfiltration and kidney injury lead to acute deterioration in residual kidney function after donor nephrectomy [6,7]. Moreover, the concentration of uremic toxins, such as indoxyl sulphate and p-cresylsulphate (which are thought to have proinflammatory effects [8,9] as well as being associated with chronic kidney disease progression [8,10], cardiovascular morbidity [11,12], and mortality [13,14]) are increased in donor serum for up to two years following nephrectomy [15].

Remote ischemic preconditioning (RIPC) has been shown to protect against ischemic reperfusion injury in various target organs, including the kidneys [16,17]. The mechanisms mediating the protective effect of RIPC have been investigated in-depth but are yet to be fully elucidated [18]. RIPC is known to exert protective effects using three different modes: a humoral, a neural, and a systemic generalized response [18]. These protective effects have been attributed to several trigger factors, which include autacoids released into the systemic circulation (nitric oxide or nitrite) [19] along with other types of signaling molecules (endocannabinoids, stromal cell-derived factor-1α, microRNA-144) [20]. These molecules activate the signaling cascade that includes protein kinase and nuclear factor kB and leads to the synthesis of protective proteins, such as manganese superoxide dismutase and inducible nitric oxide synthase [21]. This results in attenuation of endothelial dysfunction, modulation of reactive oxygen species and proinflammatory mediator release after reperfusion, and decreased cell death [21,22].

Studies on the effects of RIPC on post-transplant function in renal transplant recipients have yielded conflicting results [23,24]. Moreover, no well-designed clinical trial to date has assessed the effects of RIPC on remaining kidney function in KT donors. The anti-inflammatory effects inflicted by RIPC [25] suggest that it may prevent additional injury to the remaining kidney. In addition, ischemic preconditioning of donor organs was found to reduce ischemic reperfusion injury and improve outcomes after liver transplantation [26,27]. However, it is unclear whether RIPC can reduce ischemic reperfusion injury in the renal graft. Therefore, we hypothesized that RIPC attenuates the decrease in remnant kidney function in donors following living donor nephrectomy. Additionally, we assessed the effects of RIPC on post-transplant renal function of recipients following living donor KT.

## 2. Methods

The study was approved by the Asan Medical Center institutional review board (No. 2015-1173). All kidney donors provided informed consent. For recipients, the need for an informed consent form was waived by our ethical committee due to observational design. The randomized controlled trial registration number is KCT0002875.

All living kidney donors aged 20–60 years for elective KT were included. Donors with a history of acute kidney injury, morbid obesity, liver cirrhosis, hypertension, diabetes mellitus, hypoalbuminemia (<3.0 g/dL), hypoproteinemia (<6.0 g/dL), anemia (hemoglobin <10.6 g/dL, hematocrit <30%), or low body weight (body mass index <18 kg/m^2^) were excluded. The recipients were included according to the matched donors. Recipients were divided into the control or RIPC group according to their donor group. Two recipients in the control group were excluded from the analysis of outcomes because they were aged <8 years.

### 2.1. Donors

A total of 170 (106 related and 64 unrelated) donors were randomized 1:1 to the RIPC and control groups using a randomization table generated using R (version 3.3.1, R Foundation for Statistical Computing, Vienna, Austria) by a trial administrator who was independent of all other aspects of the trial. The surgeon, donors, and investigators were blinded to randomization.

General anesthesia in all donors and recipients was routinely induced with 4–5 mg/kg thiopental sodium and 0–50 µg fentanyl and maintained with desflurane and 50% oxygen in air with remifentanil target-controlled infusion. Muscle relaxation was induced by injecting 0.6–1.2 mg/kg rocuronium before intubation and maintained with a bolus of 10–20 mg rocuronium, as needed. All patients underwent nephrectomy by laparoscopic or hand-assisted laparoscopic surgery. After intubation, a catheter was inserted into the radial artery. When the vital signs of donors stabilized, after surgical positioning and before incision, an independent anesthesiologist (i.e., not otherwise involved in this study) performed RIPC. Briefly, a blood pressure cuff was placed on the upper arm, followed by three cycles of inflation of the cuff to 200 mmHg for 5 min and reperfusion with deflation of the cuff for 5 min. In the control group, the cuff was placed on the upper arm, but the cuff was not inflated.

The donor kidney was preserved using cold storage in histidine-tryptophan-ketoglutarate solution.

The renal function of donors was analyzed by measuring serum creatinine (sCr) concentration and estimated glomerular filtration rate (eGFR), based on sCr [28] at 1 day, 3 days, and 7 days; and 1 month, 6 months, and 12 months after donor nephrectomy. Additionally, serum cystatin C was measured at the end of surgery. The baseline eGFR in donors was measured the day before the operation.

### 2.2. Recipients

The 170 recipients were included and retrospectively evaluated by review of medical records. Kidney transplantation was performed by the local standardized protocol. The graft function of recipients was assessed after surgery by measuring sCr concentration and eGFR based on the Chronic Kidney Disease Epidemiology Collaboration (CKD-EPI) equation at 1 day, 3 days, and 7 days; and 1 month, 6 months, and 12 months after transplantation. The baseline eGFR was measured the day before the operation after dialysis.

The main immunosuppressive protocol consisted of basiliximab as an induction agent, and maintenance immunosuppressants consisted of a combination of a calcineurin inhibitor (tacrolimus or cyclosporine), mycophenolic acid, and prednisolone. As another option, recipients (*n* = 21) with immunologic risk factors (highly sensitized patients or re-transplant recipients) and those with complications due to long-term use of steroids received rabbit anti-thyomocyte globulin (Thymoglobulin^®^, Genzyme, Cambridge, MA, USA) as an induction regimen, and maintenance immunosuppression included tacrolimus, mycophenolic acid, and early steroid withdrawal in a week.

### 2.3. Intraoperative Fluid and Hemodynamic Management

For donors, Ringers Lactate or Plasmalyte was primarily administered, concomitantly with 5% albumin and mannitol. For recipients, plasma solution with 20% albumin and mannitol were used. For donors, hypotension was defined as mean blood pressure decreased below 65 mmHg and hypertension as blood pressure exceeding > 20 mmHg of baseline. For recipients, a blood pressure change of more than 20% from baseline level was diagnosed as hypotension and hypertension. If hypotension occurred, both donors and recipients were treated with inotropics or vasopressors as appropriate. In case of hypertension, nicardipine or labetalol was used. Transfusion was performed when hemoglobin level was <8 g/dL.

### 2.4. Outcomes

The primary outcome measure was the remaining kidney function post-nephrectomy in living kidney donors reflected by sCr levels and eGFR at day of discharge. Secondary outcome measurements were: in donors, incidence of chronic kidney disease 1 year after surgery, according to the Kidney Disease Improving Global Outcomes (KDIGO) criteria and in recipients, time required to a 50% decrease in baseline sCr concentration; eGFR 12 months after KT; incidence of delayed graft function, acute rejection and graft failure within the first 12 months post-transplantation. Delayed graft function was defined as a failure to <10%/day reduction in sCr concentration on 3 consecutive days during the first week after KT. Rejection was defined as any biopsy-proven rejection treated with pulsed methylprednisolone. Biopsy-proven rejection was defined as any rejection grade according to the Banff criteria [29], based on histopathological appearance of a needle core biopsy of the transplant kidney. eGFR at discharge and 1 year postoperative was adjusted by donor age and donor eGFR.

### 2.5. Statistical Analysis

Data are reported as mean ± standard deviation (SD), median (interquartile range, IQR), or mean with 95% confidence interval (CI), as appropriate. Categorical variables were compared using Pearson’s chi-square or Fisher’s exact test. Sample size was determined based on the results of a preceding study, from which the data were not shown in the final publication [30]. Briefly, we obtained the minimum eGFR (calculated by the CKD-EPI equation) during the first post-nephrectomy week from 1647 donors of KT through the preceding study. The mean ± SD of minimal eGFR during post-nephrectomy week 1 was 55 ± 11 mL/min/1.73 m^2^. The sample size was calculated based on the assumption that RIPC would increase eGFR by 10%. Based on the formula for calculating the sample size [31], 75.9 people per group would be required to detect this difference with a power of 80% and a statistical significance level of α = 0.05. Assuming a dropout rate of 10%, the desired sample size per group was 85 persons.

The statistical significance of the changes in sCr and eGFR, which were measured repeatedly according to time, was estimated using generalized mixed linear modelling to handle the dependencies in repeated measurements within the same person. All of the dependent variables at each time point after the surgery were estimated using the least square means and standard error (SE).

Risk factors for impaired kidney function in donors at discharge were investigated using multivariate logistic regression analysis. Multivariate analysis was conducted for each variable with *p* < 0.1 in the univariate analysis. All statistical analyses were performed using SPSS version 21.0 for Windows (SPSS Inc, Chicago, IL, USA), with *p*-values <0.05 defined as statistically significant.

## 3. Results

Between April 2016 and August 2017, 173 kidney donors consented to participate in this trial. After excluding three donors, 170 donors were enrolled, with 85 each randomized to the RIPC and control groups (Figure 1). Two recipients in the control group were excluded from the analysis of outcomes because they were aged <8 years.

The demographic and intraoperative characteristics of donors and recipients are summarized in Table 1 and Table 2. Age, body mass index, and American Society of Anesthesiologists physical status classifications were similar in the two groups of donors, but gender distribution differed significantly, with the proportion of men in the RIPC and control groups being 45.9% and 54.1%, respectively (*p* = 0.003). Preoperative sCr concentration of donors was higher in the control than in the RIPC group (0.8 (0.7–0.9) mg/dL vs. 0.7 (0.6–0.8) mg/dL, *p* = 0.005). Demographic characteristics did not differ significantly between the two groups of recipients. Intraoperative variables also did not differ significantly between the RIPC and the control group of donors and recipients, except that warm ischemia time in recipients was significantly shorter in the RIPC group than in the control group (25.0 (23.0–29.0) min vs. 28.0 (24.0–31.0) min, *p* < 0.01).

The amounts of each fluid used in donors and recipients are summarized in Table 1 and Table 2, respectively. Twenty-two recipients were treated with continuous administration of dobutamine, or norepinephrine as appropriate, because hypotension persisted despite bolus volume treatment. There was no significant difference in the use of inotropic or vasopressors between the treatment and control groups. None of the donors were treated with either inotropic or vasopressor agents.

Donor serum creatinine levels at discharge are depicted in Figure 2. Serum creatinine concentration was significantly increased in the control group (mean, 1.13; 95% confidence interval (CI), 1.07–1.18) compared to the RIPC group (1.01; 95% CI, 0.95–1.07) (*p* = 0.003). The proportion of donors with serum creatinine >1.4 mg/dL at discharge was higher in the control group (*n* = 17, 20.0%.) than in the RIPC group (*n* = 6, 7.1%) (*p* < 0.05). Donors with serum creatinine >1.4 mg/dL at discharge had a higher prevalence of chronic kidney disease (*n* = 6, 26.1%) than donors with normal serum creatinine level (*n* = 8, 5.4%) (*p* = 0.003) after one year. Moreover, the increase in sCr between preoperative levels and levels at discharge was greater in the control (0.35 (0.26–0.47)) than in the RIPC group (0.30 (0.22–0.42)) (*p* = 0.03).

Outcomes in donors and recipients after KT are summarized in Table 3.

RIPC, eGFR, and serum cystatin C concentrations measured 30 min after donor nephrectomy were associated with sCr level >1.4 mg/dL at discharge in donors (Table 4).

Serial changes in eGFR and sCr from donors and recipients are depicted in Figure 3. Among the donors, sCr concentrations were significantly lower in the RIPC group than in the control group on postoperative day two (*p* < 0.01), discharge day (*p* < 0.01), one month postoperatively (*p* = 0.04), and one year postoperatively (*p* = 0.02). eGFR showed a rapid increasing trend in the RIPC group in donors from postoperative day two until one month, although the between-group difference was not statistically significant. However, the linear mixed-effect modeling demonstrated a significant group-by-time interaction in terms of serial changes of eGFR (*p* < 0.01) among the donors. However, the differences in sCr between preoperative and the point where sCr was maximally increased, preoperative and discharge, or preoperative and postoperative one-year were not significant.

The male and female donors did not show any significant differences in perioperative characteristics according to treatment (Appendix A). No significant differences in serial changes of sCr or eGFR according to treatment were found between the male and female donors (Appendix A).

Among recipients, no significant differences were found in the serial changes of sCr and eGFR, and group-by-time interaction up to one year. In addition, there were no significant between-group differences in time to 50% reduction in baseline sCr concentration, or in the proportions with delayed graft function, acute rejection, and graft failure, even after adjustment for donor age and eGFR.

## 4. Discussion

This study demonstrated that RIPC attenuated the decrease in remnant kidney function in donors after donor nephrectomy. The postoperative sCr level after donor nephrectomy was statistically lower in the RIPC group, and the low sCr was maintained until one year after the operation. The increase in sCr between preoperative levels and levels at discharge was significantly greater in control group. Moreover, the proportion of kidney donors with sCr concentration >1.4 mg/dL at discharge was lower in the RIPC group. Donors with serum creatinine within normal range at discharge had a lower prevalence of chronic kidney disease (*n* = 8, 5.4%) than donors with sCr concentration >1.4 mg/dL (*n* = 6, 26.1%) after one year. Moreover, the recovery of eGFR was more rapid in the RIPC group than in the control group among donors during the first postoperative month. However, preconditioning of donor kidney grafts did not improve graft function in the recipients over one year.

RIPC was derived from ischemic preconditioning techniques and has been shown to reduce ischemic reperfusion injury, characterized by a brief period of nonlethal ischemia and reperfusion at a distant site [32]. RIPC may be a renoprotective strategy [33], reducing the incidence of contrast medium-induced acute kidney injury [34].

Donor nephrectomy may induce intrarenal structural changes and hemodynamic derangements for several days due to hyperfiltration, accompanying increases in sCr concentrations [6]. Although this is regarded as a physiological response, followed by an increase in glomerular filtration, we recently reported that a continuous increase in sCr after donor nephrectomy can lead to chronic kidney disease [30]. Similarly, kidney donors have been reported to be at greater risk of end-stage renal disease than matched healthy non-donors [4,5]. End-stage renal disease may be the long-term consequence of ongoing structural damage [35] or of the inflammatory effects of increased uremic toxins after nephrectomy [15].

We hypothesized that RIPC could reduce renal injury, followed by compensatory hyperfiltration in the remnant kidney. Our results show that RIPC was significantly effective in lowering creatinine level in donors compared to the control group not only during the acute postoperative phase, but at later times for up to one year. Although preoperative sCr was low in the RIPC group, the difference was not big and mean sCr concentrations of both groups were within normal level. In addition, eGFR level increased more rapidly during postoperative month one in the donor RIPC group, although there was no statistical significance between groups. Though we do not know the exact reason for the discrepancy between creatinine and eGFR, the small sample size might be one reason for the failure to identify statistically significant differences in the changes of eGFR between groups. Although we estimated the sample size based on a 10% increase in GFR in donors, the power of this study might not be sufficient to validate the hypothesis that RIPC can improve eGFR to a statistically significant degree in donors. The relatively short follow-up time might be another reason for the discrepancy between sCr and eGFR. Previous studies demonstrated that GFR returned to 70% of baseline by six months and continued to increase over the next three years [36]. Therefore, a longer follow-up period may be required to assess the effect of RIPC on residual renal function in living kidney donors.

Although serum cystatin C level, a marker of renal injury, was measured immediately following donor nephrectomy, no significant differences between the RIPC and control groups were found. However, a higher level of cystatin C was associated with sCr level >1.4 mg/dL, consistent with our previous finding that serum cystatin C concentration could detect partial recovery of kidney function and progression to chronic kidney disease in kidney donors [30]. Because the production of cystatin C remains constant, independent of gender, age, or muscle mass, and is sensitive only to changes in GFR [37], changes in serum cystatin C concentration allow for the early detection of deterioration in renal function. However, further studies are required to confirm that cystatin C concentration is a long-term marker of residual renal function.

In contrast to findings in KT donors, ischemic preconditioning did not improve kidney function in recipients [23,38], a finding confirmed in the present study. In our results, the RIPC group had a shorter warm ischemic time because of the higher proportion of hand-assisted laparoscopic surgeries performed in this group. After adjustment for the difference in warm ischemic time, the outcome variable did not show any significant difference. The inability of preconditioned donor kidneys to improve renal function in recipients may be due, at least in part, to the lower ability of RIPC to protect against ischemic reperfusion injury in patients at risk of lesser than higher extent of ischemic reperfusion injury [39]. Grafts from living kidney donors are exposed to shorter ischemic times than grafts from deceased donors.

Moreover, ischemic preconditioning of marginal liver grafts resulted in better transplant outcomes than ischemic preconditioning of better grafts [26], suggesting that RIPC may yield benefits for poorly conditioned kidney grafts. In addition, many studies suggest that several comorbidities and medications can interfere with the efficacy of RIPC-induced protection [32]. Preclinical studies have reported that older age and comorbid diseases, including hyperlipidemia, diabetes, and hypertension, all of which are common in KT recipients, raise the threshold for protection, requiring more robust conditioning signals [32]. In addition, sulfonylureas, which are used to treat patients with type-2 diabetes mellitus, can attenuate the conditioning effect [32]. Conversely, insulin, metformin, some statins, angiotensin-converting-enzyme inhibitors, antiplatelet agents, and opioids raise the threshold for additional benefits [32].

This study had several limitations. First, because the sample size calculation was based on a 10% increase in GFR in donors, the power of this study is not sufficient to validate the hypothesis that preconditioned grafts can improve transplantation outcomes in recipients. The statistically negative result cannot exclude the possibility of type II errors. A second limitation is that the gender distribution differed significantly between the RIPC group and the control group. The proportion of male donors was higher in the control group than in the RIPC group. Male donors have larger kidneys and greater renal reserve, which would drive findings towards null. However, there was no gender difference in the effect of RIPC, as shown in the Appendix A. Lastly, this study used a relatively short follow-up time, preventing the determination of long-term effects of RIPC on renal function in donors and recipients.

## 5. Conclusions

In conclusion, RIPC attenuated the decrease in remnant kidney function in donors after donor nephrectomy but did not improve overall renal function in kidney transplant recipients. In addition, donors with serum creatinine within normal range at discharge had lower prevalence of chronic kidney disease than donors with sCr concentration >1.4 mg/dL after one year. However, there was no improvement of overall renal function in kidney transplant recipients. While the protective mechanism of RIPC has been elucidated profoundly, clinical validation of RIPC in reducing ischemic reperfusion injury remains to be tested through a better-designed and larger study. It seems especially imperative in future studies to find the most effective intensity and timing of RIPC, and to investigate whether the effects of RIPC would be continued.

## Figures and Tables

**Figure 1 jcm-08-00713-f001:**
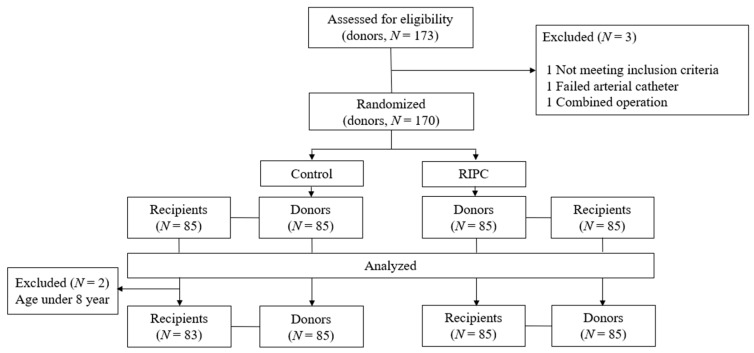
Consort diagram. Between April 2016 and August 2017, 173 donors and their recipients consented to participate in the trial. After excluding three pairs, 170 pairs of donors and recipients were enrolled, with 85 pairs randomized to each group. RIPC, remote ischemic preconditioning.

**Figure 2 jcm-08-00713-f002:**
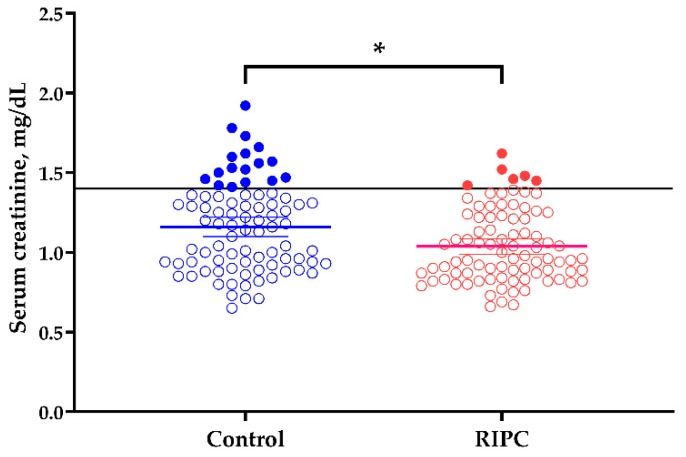
Mean serum creatinine concentration at discharge. Closed circles indicate the patients who showed high serum creatinine concentration at discharge. The black horizontal line shows the upper normal limit of serum creatinine concentration (1.4 mg/dL). Mean serum creatinine concentrations with 95% confidence interval at discharge in control (1.13 (1.07–1.18)) and RIPC (1.01 (0.95–1.07)) groups are depicted in each column (blue: control, red: RIPC). *, *p* = 0.003.

**Figure 3 jcm-08-00713-f003:**
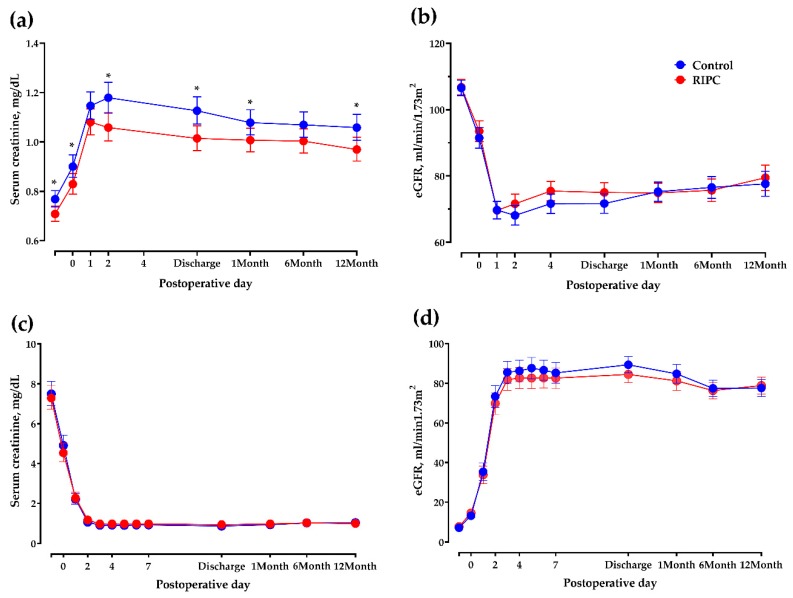
Serial changes in sCr concentration and estimated glomerular filtration rate in kidney transplant donors and recipients one year after transplantation. (**a**) Among donors, postoperative sCr concentrations were significantly higher in the control than in the RIPC group on postoperative days (PODs) zero (0.9 mg/dL, 95% CI: 0.86–0.95 mg/dL vs. 0.83 mg/dL, 95% CI: 0.79–0.87 mg/dL, *p* = 0.024), two (1.18, 95% CI: 1.12–1.24 mg/dL vs. 1.06, 95% CI: 1.00–1.12, *p* = 0.005), discharge day (1.13, 95% CI: 1.07–1.18 mg/dL vs. 1.01, 95% CI: 0.97–1.07, *p* = 0.003), one month (1.08, 95% CI: 1.03–1.13 mg/dL vs. 1.01, 95% CI: 0.96–1.06, *p* = 0.043), and 12 months (1.06, 95% CI: 1.01–1.11 mg/dL vs. 0.97, 95% CI: 0.92–1.02, *p* = 0.015). (**b**) Among donors, estimated glomerular filtration rate (eGFR) was calculated using the CKD-EPI equation. eGFR was not significantly different in the control and the RIPC groups. (**c**) Among recipients, sCr concentration did not differ in the RIPC and control groups at any time point. (**d**) Estimated glomerular filtration rate (eGFR) was calculated using the Chronic Kidney Disease Epidemiology Collaboration (CKD-EPI) equation. Among recipients, eGFR was not different between in the RIPC and control groups at any time point.

**Table 1 jcm-08-00713-t001:** Perioperative characteristics of kidney donors.

	Control Group (*n* = 85)	RIPC Group (*n* = 85)	*p*-Value
Preoperative characteristics			
Age, year	42.0 (33.0–49.0)	44.0 (37.0–51.0)	0.33
Males, *n* (%)	46 (54.1)	26 (30.5)	<0.01
Body mass index, kg/m2	24.0 ± 2.8	23.7 ± 2.9	0.45
sCr, mg/dL	0.8 (0.7–0.9)	0.7 (0.6–0.8)	<0.01
eGFR, mL/min/1.73 m2	106.6 ± 11.1	106.8 ± 10.6	0.90
ASA PS 1/2	60/25	59/26	1.00
Intraoperative characteristics			
Operation time	212.0 (190.0–235.0)	205.0 (183.0–225.0)	0.09
HALS, *n* (%)	66 (77.7)	75 (88.2)	0.10
Kidney (right), *n* (%)	34 (40.0)	36 (42.4)	0.88
Plasmalyte, mL	1302.1 ± 547.3	1205.9 ± 534.1	0.25
Transfusion of RBC, *n* (%)	0 (0)	1 (1.2)	1.00
20% Mannitol, mL	160.0 (140.0–190.0)	150.0 (14.0.0–175.0)	0.33
Urine output, mL	270.0 (220.0–420.0)	290.0 (190.0–400.0)	0.37
Mean blood pressure, mmHg			
Before RIPC	76.0 (71.0–89.0)	78.0 (71.0–89.0)	0.63
After RIPC	84.2 ± 9.6	85.4 ± 12.8	0.51
Cystatin C, mg/L	0.9 ± 0.1	0.9 ± 0.2	0.39

RIPC, remote ischemic preconditioning; sCr, serum creatinine; ASA PS, American Society of Anesthesiologists physical status; eGFR, estimated glomerular filtration rate based on the Chronic Kidney Disease Epidemiology Collaboration (CKD-EPI) equation; HALS, hand-assisted laparoscopic surgery; RBC, red blood cells. Data are expressed as mean ± standard deviation (SD), median and interquartile range, or number (%), as appropriate.

**Table 2 jcm-08-00713-t002:** Perioperative characteristics of kidney recipients.

	Control Group (*n* = 83)	RIPC Group (*n* = 85)	*p*-Value
Preoperative characteristics			
Age	45.9 ± 13.4	47.6 ± 10.0	0.34
Males, *n* (%)	43 (50.6)	46 (54.1)	0.65
Body mass index, kg/m^2^	22.6 ± 3.6	22.8 ± 3.3	0.68
sCr, mg/dL	7.9 (5.7–9.2)	7.5 (6.0–9.1)	0.53
eGFR, mL/min/1.73 m^2^	7.0 (5.0–9.0)	7.0 (5.0–9.0)	0.41
Duration of chronic renal failure, months	72.9 ± 84.1	70.8 ± 58.4	0.22
Dialysis, *n* (%)	67 (78.8)	64 (75.30)	0.58
Second transplantation, *n* (%)	9 (10.8)	6 (7.1)	0.39
ABO incompatibility, *n* (%)	28 (33.7)	25 (29.4)	0.55
Hypertension, *n* (%)	76 (89.4)	70 (82.4)	0.19
Cerebrovascular accident, *n* (%)	2 (2.4)	0 (0)	0.50
Diabetes mellitus, *n* (%)	18 (21.2)	22 (25.9)	0.47
Ischemic heart disease, *n* (%)	4 (4.7)	2 (2.4)	0.68
Intraoperative characteristics			
Duration of operation, min	267.0 (252.0–302.5)	267.0 (230.0–310.0)	0.51
Plasmalyte, mL	950 (700–1300)	1000 (700–1260)	0.83
Transfusion of RBC, *n* (%)	5 (6.0)	3 (3.5)	0.69
20% Mannitol, mL	150 (125–170)	150 (125–175)	0.85
Cold ischemic time, min	41.0 (26.5–56.0)	37.0 (22.0–55.0)	0.23
Warm ischemic time, min	28.0 (24.0–31.0)	25.0 (23.0–29.0)	<0.01
Lowest mean blood pressure, mmHg			
Before reperfusion	78.0 (70.0–83.0)	79.3 (71.0–87.3)	0.30
After reperfusion	84.8 ± 11.4	87.0 ± 11.1	0.20
Intraoperative inotropic use, *n* (%)	12	10	0.77
Dobutamine	6	9	-
Norepinephrine	6	1	-
Urine output, mL	430 (260–745)	440 (300–680)	0.80

RIPC, remote ischemic preconditioning; sCr, serum creatinine; eGFR, estimated glomerular filtration rate based on the Chronic Kidney Disease Epidemiology Collaboration (CKD-EPI) equation; ABO incompatibility, ABO incompatibility between recipients and donors; RBC, red blood cells. Data are expressed as mean ± standard deviation (SD), median and interquartile range, or number (%), as appropriate.

**Table 3 jcm-08-00713-t003:** Outcomes in kidney transplant donors and recipients.

**Donors**	**Control Group (*n* = 85)**	**RIPC Group (*n* = 85)**	***p*-Value**
Progression to chronic kidney disease, *n* (%)	6 (7.1)	8 (9.4)	0.78
Proportion with sCr ≥ 1.4 mg/dL at discharge, *n* (%)	17 (20.0)	6 (7.1)	<0.05
**Recipients**	**Control Group (*n* = 83)**	**Treated Group (*n* = 85)**	***p*-Value**
Time to 50% decrease in sCr, days	1.1 ± 0.6	1.4 ± 3.2	0.58
Delayed graft function, *n* (%)	0 (0.0)	2 (2.4)	0.49
Acute rejection, *n* (%)	5 (6.0)	5 (5.9)	1.00
Graft failure, *n* (%)	1 (1.2)	0 (0.0)	0.99
eGFR at discharge, mL/min/1.73 m^2^	87.9 ± 17.7	83.9 ± 19.4	0.18
Mean difference ^1^	−3.23 (−8.62–2.16)		0.14
eGFR at postoperative 1 year, mL/min/1.73 m^2^	77.0 ± 24.5	79.5 ± 18.2	0.49
Mean difference ^1^	3.45 (−3.11–10.01)		0.29

RIPC, remote ischemic preconditioning; sCr, serum creatinine; eGFR, estimated glomerular filtration rate. eGFR was calculated using the Chronic Kidney Disease Epidemiology Collaboration (CKD-EPI) equation. ^1^ Difference in means (95% CI) adjusted for warm ischemic time and preoperative serum creatinine, age and baseline eGFR of donor.

**Table 4 jcm-08-00713-t004:** Predictors of sCr >1.4 mg/dL at the time of discharge.

	Univariate			Multivariate		
	Odds Ratio	95% CI	*p*-Value	Odds Ratio	95% CI	*p*-Value
RIPC	0.30	0.1–0.78	0.02	0.31	0.1–0.88	0.04
Age	0.95	0.91–0.99	0.02			
ASA PS	0.80	0.27–2.07	0.66			
Body mass index	1.17	1.00–1.37	0.05			
Operation time	1.01	1.00–1.03	0.02			
Operation site	1.10	0.45–2.80	0.83			
Systolic blood pressure	1.03	1.00–1.07	0.07			
Albumin	2.56	0.52–13.04	0.25			
eGFR	0.95	0.91–0.99	0.01	0.94	0.90–0.99	<0.01
Cys C, mg/L	1.91	1.39–2.75	<0.01	2.17	1.44–3.44	<0.01

RIPC, remote ischemic preconditioning; ASA PS, American Society of Anesthesiologists physical status classification; eGFR, estimated glomerular filtration rate; CysC, cystatin C within 24 h after surgery.

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
