# Peer review of "Impact of Remote Ischemic Preconditioning Conducted in Living Kidney Donors on Renal Function in Donors and Recipients Following Living Donor Kidney Transplantation: A Randomized Clinical Trial"

_jcm, 2019, doi:10.3390/jcm8050713_

Reviewer 1 Report

Bang and colleagues present the results of a randomised controlled trial on the effect of remote ischemic preconditioning in living kidney donors on renal function in donors and recipients. Although this subject is of interest to both the anaesthesia and transplant community I have some serious concerns and comments:

First of all ideally randomised controlled trials are reported following the CONSORT 2010 guidelines. The study presented by Bang and colleagues does not follow this guidelines. Furthermore the last patient was included August 2017. The study, however, was only registered in the CRIS registration May 18 2018. Can  the authors please explain why they didn’t register this study before the inclusion started?

Overall the manuscript is rather unclear and therefore difficult to read.

Major comments:

Primary outcome:

Normally a RCT has one or a combined primary outcome measure on which the sample size calculation is based. In this study the outcome section is somewhat confusing. Please specify primary and secondary outcomes clear in the outcome section

Regarding the intraoperative fluid and hemodynamic management in both donor and recipient:

Please describe how this was managed and report this data in the result section. What was the hypotension/hypertension threshold, what was done when blood pressure dropped below/raised above this threshold? What kind of fluid was administered? What was the amount of blood loss in both groups.

Storage technique of the kidney:

What was the storage technique of the kidney, cold or machine perfusion or were both techniques randomly used

Kidney function donor

How was the eGFR in the donor calculated? MDRD, CKD-EPI, why was sCR or cystatin C used. Or were they both used in all patients…In a study by Shlipak et al., in 2013  42% of persons with a creatinine-based eGFR of 45 to 59 ml per minute per 1.73 m2 had a cystatin C–based eGFR of more than 60 ml per minute per 1.73 m2 so both calculations are not interchangeable

Shlipak MG, Matsushita K, Ärnlöv J, et al. Cystatin C versus creatinine in determining risk based on kidney function. N Engl J Med. 2013;369:932–43

When was baseline eGFr in the donors measured? In our center we perform a measured GFR including a forced GFR with dopamine.

Why was DGF in the recipients based on 3 different definitions. Since al 3 definitions have their own strengths and weaknesses they are not interchangeable. Please use one definition. I am aware of the fact that a dialysis based definition is difficult in pre-emptively transplanted patients however the functional DGF defintion,  failure of serum creatinine level to decrease by at least 10% on 3 consecutive days during the first post-operative week is usable. Moore and colleagues showed that fDGF is independently associated with reduced death-censored graft survival in contrast to DGF based on the dialysis definition and suggest a superiority of this definition over the dialysis based definition. In patients with excellent early graft function when optimal graft function had already been achieved by day 2  you can prevent misclassification, by not classify them as fDGF

Moore J, Shabir S, Chand S et al: Assessing and comparing rival definitions of delayed renal allograft function for predicting subsequent graft failure. Transplantation, 2010; 90: 1113–16

Rejection and Graft function

Since acute rejection and graft failure are mentioned as secondary outcome measures please take risk factors for these outcomes  such as HLA-mismatches, immunosuppressive protocol, panel reactive antibodies and donor specific antibodies into account.

How was the time required to a 50% decrease in baseline sCr estimated? We used this outcome in one of our own studies and it is nog an easy outcome parameter warranting multiple sCr measurements on various time points and lineair or logarithmic modelling mathematical modelling.

Krogstrup NV, Bibby BM, Aulbjerg C, Jespersen B, Birn H.A new method of modelling early plasma creatinine changes predicts 1-year graft function after kidney transplantation. Scand J Clin Lab Invest. 2016 Jul;76(4):319-23

Regarding the endpoint sCr>1.4 mg/dl at day of discharge. This says little about the actual sCr levels of these patients. Theoretically it could be that in the RIPC group these levels were actually higher than in control. Better report sCr levels at discharge with 95% CI  as you do in Figure 3 a. One can ask whether the differences in sCr between the groups are clinically relevant.

Regarding the Multivariate analysis: Since there are only 6 + 17 =23“events” namely an sCr >1.4 mg/dl at day of discharge you can only adjusted the model in the multivariate regression analysis for no more than 2 or 3 factors. It is unclear to my how the MRA was performed, which approach was used? The 95%CI  of RIPC seems rather large.

What was the relation of the kidney donors to the recipients? Please report the amount of related and unrelated donors.

In the introduction the concept of RIC is described shortly and to my opinion to minimal. This however could be more accurate. I would refer the authors to the following review about the molecular aspects of (R)IC:

Kierulf-Lassen C, Nieuwenhuijs-Moeke GJ, Krogstrup NV, Oltean M, Jespersen B, Dor FJ. Molecular Mechanisms of Renal Ischemic Conditioning Strategies. Eur Surg Res. 2015;55(3):151-83

Table 1: please make separate section of recipients

Author Response

Point 1: First of all ideally randomised controlled trials are reported following the CONSORT 2010 guidelines. The study presented by Bang and colleagues does not follow this guidelines. Furthermore the last patient was included August 2017. The study, however, was only registered in the CRIS registration May 18 2018. Can the authors please explain why they didn’t register this study before the inclusion started?

Response 1: We apologize because we did not follow the CONSORT guidelines. As this study was our first randomized controlled trial, we did not know that we should register and enter the information of the study on the CRIS registration before the inclusion started. We performed the CRIS registration as soon as possible when we became aware of.

Point 2: Overall the manuscript is rather unclear and therefore difficult to read.  

Response 2: We apologize for the inconvenience. Before submission of the manuscript, we had the manuscript checked by a professional English editing service. As the reviewer had such inconvenience, we had the manuscript checked again by a professional English editing service. 

Major comments:

Point 3: Primary outcome:  Normally a RCT has one or a combined primary outcome measure on which the sample size calculation is based. In this study the outcome section is somewhat confusing. Please specify primary and secondary outcomes clear in the outcome section.

Response 3: Thank you for your suggestion. We clarify our primary and secondary outcome measures in donors and recipients. Now it reads,

“Primary outcome was post-nephrectomy and post-transplantation renal function according to RIPC in donors and recipients. Renal function in donors and recipients was assessed with the sCr concentrations and eGFR at discharge day as primary outcome measurements. Secondary outcome measurement in donors was the proportion with chronic kidney disease, defined according to Kidney Disease Improving Global Outcomes (KDIGO) criteria, 1 year after surgery. For recipients, time required for a 50% decrease in baseline sCr concentration; eGFR 12 months after KT; the proportion with delayed graft function; and the proportions with acute rejection and graft failure during the first 12 months were investigated as secondary outcome measurements. Delayed graft function was defined as a<10%/day reduction in sCr concentration on 3 consecutive days during the first week after KT.”

Point 4: Regarding the intraoperative fluid and hemodynamic management in both donor and recipient:  Please describe how this was managed and report this data in the result section. What was the hypotension/hypertension threshold, what was done when blood pressure dropped below/raised above this threshold? What kind of fluid was administered? What was the amount of blood loss in both groups.

Response 4: We apologize for the incomplete description. We did not count blood loss intraoperatively, because estimation of blood loss is not only a difficult task but also inaccurate. We performed transfusion when hemoglobin level was<8 g/dL. We added the subsection “2.3. Intraoperative fluid and hemodynamic management” in the methods section and summarized this information therein. Now it reads,

In methods section: “For donors, Hartman solution or plasma solution was primarily administered, concomitantly with 5% albumin and mannitol. For recipients, plasma solution with 20% albumin and mannitol were used. For donors, hypotension was defined as mean blood pressure decreased below 65 mmHg and hypertension as blood pressure exceeding > 20 mmHg of baseline. For recipients, blood pressure change of more than 20% from baseline level was diagnosed as hypotension and hypertension. Both donors and recipients were treated with inotropic or vasopressors bolus administration for hypotension, and nicardipine or labetalol was used to treat hypertension. Transfusion was performed when hemoglobin level was < 8 g/dL.

In the results; “The amounts of each fluid used in donors and recipients are summarized in Tables 1 and 2, respectively. Twenty-two recipients were treated with continuous administration of dobutamine, or norepinephrine as appropriate, because hypotension persisted despite bolus treatment. There was no significant difference in the use or inotropic or vasopressors between treatment and control groups. None of the donors was administered with either inotropic or vasopressor agent. Donor serum creatinine levels at discharge are depicted in Figure 2. More patients (n=6, 26.1%) with serum creatinine > 1.4 mg/dL at discharge than those with normal serum creatinine concentration (n=8, 5.4%) got chronic kidney disease after 1 year (P=0.003).”

Point 5: Storage technique of the kidney:  What was the storage technique of the kidney, cold or machine perfusion or were both techniques randomly use. 

Response 5: We used the cold storage technique with histidine-tryptophan-ketoglutarate solution. We added a sentence describing this in the Methods section.

Point 6: Kidney function donor:

How was the eGFR in the donor calculated? MDRD, CKD-EPI, why was sCR or cystatin C used. Or were they both used in all patients…In a study by Shlipak et al., in 2013 42% of persons with a creatinine-based eGFR of 45 to 59 ml per minute per 1.73 m2 had a cystatin C–based eGFR of more than 60 ml per minute per 1.73 m2 so both calculations are not interchangeable

   Shlipak MG, Matsushita K, Ärnlöv J, et al. Cystatin C versus creatinine in determining risk based on kidney function. N Engl J Med. 2013;369:932–43

When was baseline eGFr in the donors measured? In our center we perform a measured GFR including a forced GFR with dopamine.

Response 6: We used eGFR-based sCr using the CKD-EPI equation to evaluate renal function in donors and recipients. Cystatin C was measured only in donors. The baseline eGFR in donors was measured the day before the operation. In our center, we do not measure a forced GFR with dopamine.

Point 7: Why was DGF in the recipients based on 3 different definitions. Since al 3 definitions have their own strengths and weaknesses they are not interchangeable. Please use one definition. I am aware of the fact that a dialysis based definition is difficult in pre-emptively transplanted patients however the functional DGF defintion, failure of serum creatinine level to decrease by at least 10% on 3 consecutive days during the first post-operative week is usable. Moore and colleagues showed that fDGF is independently associated with reduced death-censored graft survival in contrast to DGF based on the dialysis definition and suggest a superiority of this definition over the dialysis based definition. In patients with excellent early graft function when optimal graft function had already been achieved by day 2 you can prevent misclassification, by not classify them as fDGF

    Moore J, Shabir S, Chand S et al: Assessing and comparing rival definitions of delayed renal allograft function for predicting subsequent graft failure. Transplantation, 2010; 90: 1113–16

Response 7: Thank you for kind suggestion. We changed the definition of DGF as per your suggestion. However, the number of the patients with DGF had not changed. Now it reads,

“Delayed graft function was defined as a<10%/day reduction in sCr concentration on 3 consecutive days during the first week after KT.”

Point 8: Rejection and Graft function:   Since acute rejection and graft failure are mentioned as secondary outcome measures please take risk factors for these outcomes such as HLA-mismatches, immunosuppressive protocol, panel reactive antibodies and donor specific antibodies into account.

Response 8: Thank you for your kind suggestion. Our primary aim was to assess the influence of RIPC on renal function in donors and recipients. Our results also showed that RIPC did not affect incidence of the acute rejection and graft failure. Therefore, investigating further risk factors of acute rejection or graft failure does not meet with our aim. We will investigate those factors in the future study.

Point 9: How was the time required to a 50% decrease in baseline sCr estimated? We used this outcome in one of our own studies and it is not an easy outcome parameter warranting multiple sCr measurements on various time points and linear or logarithmic modelling mathematical modelling.

   Krogstrup NV, Bibby BM, Aulbjerg C, Jespersen B, Birn H.A new method of modelling early plasma creatinine changes predicts 1-year graft function after kidney transplantation. Scand J Clin Lab Invest. 2016 Jul;76(4):319-23

Response 9: Thank you for your advice. In our center, sCr was measured twice daily on postoperative days 0 and 1, then daily until discharge. After discharge, serum creatinine was measured once per 3 weeks for first 3 months, then once every 3 months until 1 year. We identified the time required for a 50% decrease in baseline sCr estimated by graphical inspection, then compared the time between control and RIPC groups. Our result was similar with the result you mentioned.

Point 10: Regarding the endpoint sCr>1.4 mg/dl at day of discharge. This says little about the actual sCr levels of these patients. Theoretically it could be that in the RIPC group these levels were actually higher than in control. Better report sCr levels at discharge with 95% CI as you do in Figure 3 a. One can ask whether the differences in sCr between the groups are clinically relevant.

Response 10: Thank you for bright suggestion. We changed Figure 2, which showed the mean serum creatinine concentration with 95% CI at discharge. You mentioned clinical relevance of the difference in serum creatine. However, it is well known that minimal increase in serum creatinine increases the risk of acute kidney injury or CKD. Our results also showed that more patients (6, 26.1%) with serum creatinine >1.4 mg/dL at discharge than the patients with normal serum creatinine concentration (8, 5.4%) got CKD after 1 year (P=0.003). We summarized these results in the Results section.

Point 11: Regarding the Multivariate analysis: Since there are only 6 + 17 =23“events” namely an sCr >1.4 mg/dl at day of discharge you can only adjusted the model in the multivariate regression analysis for no more than 2 or 3 factors. It is unclear to my how the MRA was performed, which approach was used? The 95%CI of RIPC seems rather large.

Response 11: Thank you for your thoughtful point. We consulted with our statistician to perform multiple logistic regression analysis by backward elimination. As we found correlation between age and eGFR (correlation coefficient -0.6), we performed the multiple logistic regression again after excluding the variable “age”. RIPC, eGFR, and Cystatin C were associated with increased serum creatinine concentration (1.4 mg/dL) at discharge, which met the criteria for no more than 3 factors. The revised Table 4 is shown in the Results section.

Point 12: What was the relation of the kidney donors to the recipients? Please report the amount of related and unrelated donors.

Response 12: There were 106 related and 64 unrelated donors. We mentioned this information in the revised Methods section.

Point 13: In the introduction the concept of RIC is described shortly and to my opinion to minimal. This however could be more accurate. I would refer the authors to the following review about the molecular aspects of (R)IC:

Kierulf-Lassen C, Nieuwenhuijs-Moeke GJ, Krogstrup NV, Oltean M, Jespersen B, Dor FJ. Molecular Mechanisms of Renal Ischemic Conditioning Strategies. Eur Surg Res. 2015;55(3):151-83

Response 13: Thank you for thoughtful suggestion. We referenced the article you suggested, and we added more description on the molecular mechanism of RIPC. Now it reads as,

“The mechanisms mediating protective effect of RIPC have been investigated in-depth but are yet to be fully elucidated [Kierulf-Lassen, C. et al., Eur. Surg. Res 2015, 55, 151-83]. RIPC is known to exert protective effects using three different modes: a humoral, a neural, and a systemic generalized response [Kierulf-Lassen, C. Et al, Eur. Surg. Res 2015, 55, 151-83]. These protective effects have been attributed to several trigger factors, which include autacoids released into the systemic circulation (nitric oxide or nitrite)[ Rasaaf, T et al.,Circ. Res. 2014, 114, 1601-10] and other types of signaling molecules (endocannabinoids, stromal cell-derived factor-1α, microRNA-144) [Pickard, J.M., et al.,Basic Res. Cardiol. 2015, 110, 453]. These molecules activate the signaling cascade that includes protein kinase and nuclear factor kB and leads to the synthesis of protective proteins, such as manganese superoxide dismutase and inducible nitric oxide synthase [Carini, R, et al., Gastroenterology 2003, 125, 1480-91].”

Point 14: Table 1: please make separate section of recipients

Response 14: Thank you for your advice. We made split the table into Table 1 and 2 in the revised manuscript.

Reviewer 2 Report

Authors present interesting paper in which they have studied whether RIPC performed in living kidney donors could improve residual renal function in donors and outcomes in recipients following kidney transplantation. This approach truly is original and thus could be pointed out as the most important issue of this work. Hypotheses are clear and study design is such that it may give answers to the questions behind these hypotheses. Paper is well written and easy to read. I have some questions/concerns.

Methods:
• 170 donor recipient pair were selected according to well described exclusion criteria. Randomization is also well explained. However, how were these 170 pair selected?
• Intervention is well explained. Have the authors performed/considered any biochemical measurements that could show that this intervention has been effective to initiate mechanisms that are suggested to be behind the positive effects of RIPC?
• Some donor operations were performed laparoscopically and some with hand assisted laparoscopic style. Data about these is lacking and the effect of this difference should be discussed at least.

Results:
• Preoperative sCr concentration was higher in the control than in the RIPC group. This fact may at least partly explain the main results. Issue should be discussed more. In addition, warm ischemia time in recipients was significantly shorter in the RIPC group than in the control group. What is the impact of this. Have the authors adjust these diffenrences prior to statistical analyses?
• Figure 2 gives no additional information.

Discussion:

• Authors merely summarize the main results at the end of discussion, but could they give readers any suggestion concerning RIPC in clinical work?

Author Response

Authors present interesting paper in which they have studied whether RIPC performed in living kidney donors could improve residual renal function in donors and outcomes in recipients following kidney transplantation. This approach truly is original and thus could be pointed out as the most important issue of this work. Hypotheses are clear and study design is such that it may give answers to the questions behind these hypotheses. Paper is well written and easy to read. I have some questions/concerns.

Methods:
Point 1: 170 donor recipient pair were selected according to well described exclusion criteria. Randomization is also well explained. However, how were these 170 pair selected?

Response 1:A total of 170 donors were assessed for eligibility and randomly assigned to either the RIPC group (n=85) or the control group (n=85). For the recipients assigned to the selected donors, the medical records of 170 recipients paired to the selected donors were retrospectively analyzed, and two recipients in the control group were excluded from the analysis because they were aged<8 years.

Point 2: Intervention is well explained. Have the authors performed/considered any biochemical measurements that could show that this intervention has been effective to initiate mechanisms that are suggested to be behind the positive effects of RIPC?

Response 2: We measured cystatin C level in donors after nephrectomy. Although cystatin C level was associated with higher serum creatinine at discharge, mean cystatin C concentration did not differ between control and RIPC groups. We plan to investigate the mechanism of RIPC in detail in a future study.

Point 3: Some donor operations were performed laparoscopically and some with hand assisted laparoscopic style. Data about these is lacking and the effect of this difference should be discussed at least.

Response 3: Thank you for your opinion. One hundred forty-one (141, 82.9%) patients got hand-assisted laparoscopic surgery (HALS) in our study. While more patients in the RIPC group (n= 75, 88.2%) underwent HALS than in the control group (n=66, 77.7%), there was no significant difference. Warm ischemic time was longer in patients with the traditional laparoscopic nephrectomy (29.0 [25.0–35.0] minutes) than in patients with HALS (25.0 [23.0–30.0] minutes), as in a previous study [International Journal of Surgery 40 (2017) 83-90]. The patients in the RIPC group might have shown shorter warm ischemic time due to the influence of different type of surgery. However, it is known that no significant differences were found between the two types of nephrectomy in terms of graft survival or intraoperative complication rates [International Journal of Surgery 40 (2017) 83-90]. Therefore, effects of difference in type of surgery would be expected to be limited, but the authors believe that more through study is required. We added this result in Table 1 in the Results section and Discussion section.

Results:
Point 4: Preoperative sCr concentration was higher in the control than in the RIPC group. This fact may at least partly explain the main results. Issue should be discussed more. In addition, warm ischemia time in recipients was significantly shorter in the RIPC group than in the control group. What is the impact of this. Have the authors adjust these differences prior to statistical analyses?

Response 4: Thank you for your advice. Although preoperative serum creatinine was lower in RIPC group, the difference was not big, and means of both groups were within normal limit. Therefore, we expected that the influence of difference in serum creatinine would not be great. We added the following comment in the discussion section:

“Although preoperative sCr was low in RIPC group, the difference was not big, and mean sCr concentrations of both groups were within normal level.”

We investigated the difference of outcome variables between two groups after adjusting the warm ischemic time. After this adjustment, the results were not changed. The results are summarized in Table 3.

Point 5: Figure 2 gives no additional information.

Response 5: Thank you for helpful suggestion. We changed Figure 2, with the revised version showing the mean serum creatinine concentration with 95% CI at discharge.

Discussion:

Point 6: Authors merely summarize the main results at the end of discussion, but could they give readers any suggestion concerning RIPC in clinical work?

Response 6: Thank you for kind suggestion. We added a comment at the end of discussion, suggesting more investigation:

“While the protective mechanism of RIPC has been elucidated profoundly, clinical validation of RIPC in reducing ischemic reperfusion injury remains to be tested through a better-designed and larger study. Especially, it seems imperative in the future study to find the most effective intensity and timing of RIPC, and to investigate whether the effects of RIPC would be continued.”

Round  2

Reviewer 1 Report

The authors have to be complimented about the changes they have made to the manuscript in such a short period of time. The manuscript has improved significantly. However I still have somne concerns about outcome variables. As stated in the result section:

Preoperative sCr concentration was higher in the control than in the RIPC  group (0.8 [0.7–0.9] mg/dL vs. 0.7 [0.6–0.8] mg/dL, p=0.005).

and

Serum creatinine  concentration was significantly increased in control group (mean, 1.13; 95% confidence interval [CI], 1.07-1.18) than RIPC group (1.01; 95% CI, 0.95-1.07)(P = 0.003).

To my opinion the authors have overlooked the fact that the control group already started with higher sCr….So the delta creat in each group would be more of interest….otherwise you’ll end up with a selection bias. Did you look at the delta creat within patients?? Did this differ between groups? Based on these data I cannot agree with the first sentence in the discussion and the conclusion of the study.

Furthermore there are some other points that need clarification or improvement…

Introduction

1) Last paragraph of introduction.

Studies on the effects of RIPC on post-transplant function in renal transplant recipients have yielded conflicting results [23,24]. Moreover, no well-designed clinical trial to date has assessed the  effects of RIPC on remaining kidney function in KT donors. The anti-inflammatory effects inflicted byof RIPC [25] suggest that it may prevent additional  injury in the remainingnant donor kidney. In addition, ischemic preconditioning of donor  organs was found to reduce ischemic reperfusion injury and improve outcomes after liver transplantation [26,27]. However, it is unclear whether RIPC can reduce ischemic reperfusion injury  in the renal graft. Therefore, this studywe hypothesized that RIPC may attenuates the decrease in remnant  kidney function in donors following living donor nephrectomy. Additionally, this studywe assessed  the effects of RIPC on post-transplant renal function of recipients following living donor KT

Methods section

2) Please follow CONSORT guidelines. State clear inclusion and exclusion criteria.

3) It is unclear to me what the authors mean by: One of the tables was delivered to an anesthesiologist who was….

4) In the control group, the cuff… but the pressure was not increased. Please replace by: the cuff was not inflated.

5) Donor kidney was preserved using the simple cold storage technique with histidine-tryptophan-ketoglutarate solution. Please remove the simple and technique. What company manufactures the HTK used?

6) Cystatin C was measured only in donors at the end of operation. Remove only and replace by: Additionally serum cystatin C was measured at the end of surgery

7) What is the baseline sample time point both in donors and recipients? The authors have written it in their response but still cannot find it in the manuscript. Were dialysis patients dialysed before or after this baseline

8) please mention immunosuppressive protocol in short.

9) The 170 recipients were included according to the matched donors. Recipients were divided  into control or RIPC group according to their donor group, and retrospectively evaluated by review  of medical records. Two recipients in the control group were excluded from the analysis of outcomes  because they were aged<8 years. The donor kidney was inserted into the iliac fossa of the recipient. The iliac artery was mobilized, and about 3 cm was prepared for anastomosis. The external iliac vein  was also mobilized for end-to-side anastomosis by dividing the lymphatics overlying this vein, with  or without dividing the internal iliac vein at its origin. The donor renal vein was anastomosed to the  external iliac vein, and the donor artery was anastomosed to the recipient common or external iliac  artery in an end-to-side manner. The ureter was spatulated, and an end-to-side anastomosis was  completed to the bladder mucosa.

To my opinion the red part can be replaced by: kidney transplantation was performed by local standardized protocol. What do the authors mean with spatulated?? Don’t they mean cannulated?

10)

For donors, Hartman solution or plasma solution was primarily administered, concomitantly  with 5% albumin and mannitol. For recipients, plasma solution with 20% albumin and mannitol  were used. For donors, hypotension was defined as mean blood pressure decreased below 65  mmHg and hypertension as blood pressure exceeding > 20 mmHg of baseline. For recipients, blood  pressure change of more than 20% from baseline level was diagnosed as hypotension and  hypertension. Both donors and recipients were treated with inotropic or vasopressors bolus  administration for hypotension, and nicardipine or labetalol was used to treat hypertension. Transfusion was performed when hemoglobin level was < 8 g/dL.

Hartman solution replace by Ringers Lactate

plasma solution, what do you mean with this? Plasmalyte?

Suggestion to replace the red sentence If hypotension occurred, both donors and recipients were treated with inotropics or vasopressors… (Which ones?????). In case of hypertension nicardipine or labetolol was used.

Was the first step in case of hypo/hypertension not to adjust the depth of anesthesia/analgesia if possible??

11)

Primary outcome was post-nephrectomy and post-transplantation renal function according to  RIPC in donors and recipients. Renal function in donors and recipients was assessed with the sCr  concentrations and eGFR at discharge day as primary outcome measurements. Secondary outcome  measurement in donors was the proportion with chronic kidney disease, defined according to  Kidney Disease Improving Global Outcomes (KDIGO) criteria, 1 year after surgery. For recipients, time required for a 50% decrease in baseline sCr concentration; eGFR 12 months after KT; the proportion with delayed graft function; and the proportions with acute rejection and graft failure  during the first 12 months were investigated as secondary outcome measurements. Delayed graft  function was defined as a<10%/day reduction in sCr concentration on 3 consecutive days during the  first week after KT. Rejection was defined as any biopsy-proven or biochemical rejection treated  with pulsed methylprednisolone. eGFR at discharge and 1 year postoperative was adjusted by  donor age and donor eGFR.

Please replace by:

Primary outcome measure was the remaining kidney function post-nephrectomy in living kidney donors reflected by sCr  levels and eGFR at day of discharge. Secondary outcome  measurements were: in donors, incidence of chronic kidney disease 1 year after surgery, according to the Kidney Disease Improving Global Outcomes (KDIGO) criteria and in recipients time required to a 50% decrease in baseline sCr concentration; eGFR 12 months after KT; incidence of delayed graft function, acute rejection and graft failure the first 12 months post transplantation. Delayed graft  function was defined as a failure to<10%/day reduction in sCr concentration on 3 consecutive days during the  first week after KT. Rejection was defined as any biopsy-proven or biochemical rejection treated  with pulsed methylprednisolone. eGFR at discharge and 1 year postoperative was adjusted by  donor age and donor eGFR.

12)

Delayed graft  function was defined as a<10%/day reduction in sCr concentration on 3 consecutive days during the  first week after KT. How did you handle the patients with excellent function on day 2 which did not show further reductions on day 3…..

13)

Rejection was defined as any biopsy-proven or biochemical rejection. What do you mean with biochemical rejection. Normally in KT trials acute rejection is defined as biopsy proven and treated…

14)

eGFR at discharge and 1 year postoperative was adjusted by donor age and donor eGFR. What do you mean with this… eGFR in recipients? If so how was this performed???

15)

time required to a 50% decrease in baseline sCr concentration…

When was baseline sCr measured? Was this the same timepoint for all patients? Were dialysis patients dialyzed before or after.. This significantly influences the outcome parameter…

The authors state: We identified the time required for a 50% decrease in baseline sCr estimated by graphical inspection. This is rather vague to me please clarify.

16)

Regarding the sample size calculation: Sample size was determined as described previously [29]. Briefly, that study showed that the average eGFR calculated by the CKD-EPI equation on postoperative day 7 was 55 ± 11 mL/min/1.73 m2.

I cannot find these data in the article you are referring to…..

17)

Please describe how the Multivariate analysis was performed

18)

Flowchart of the trial……This is called a consort diagram.

19) Table 1

Operation site replace by: kidney

Plasma solution. Do you mean balanced cristalloids?

Levophed. Please replace by noradrenaline/norepinephrine

Author Response

Point 1. Preoperative sCr concentration was higher in the control than in the RIPC group (0.8 [0.7–0.9] mg/dL vs. 0.7 [0.6–0.8] mg/dL, p=0.005).

and

Serum creatinine concentration was significantly increased in control group (mean, 1.13; 95% confidence interval [CI], 1.07-1.18) than RIPC group (1.01; 95% CI, 0.95-1.07) (P = 0.003).

To my opinion the authors have overlooked the fact that the control group already started with higher sCr….So the delta creat in each group would be more of interest….otherwise you’ll end up with a selection bias. Did you look at the delta creat within patients?? Did this differ between groups? Based on these data I cannot agree with the first sentence in the discussion and the conclusion of the study.

Response 1. Thank you for your suggestion. According to your suggestion, we compared the increase in sCr between preoperative and at discharge and found out that RIPC attenuated the postoperative increase in sCr in RIPC group. The difference in sCr between preoperative and discharge was greater in control group than RIPC [control 0.35 (0.26-0.47) vs RIPC 0.30 (0.22-0.42), p=0.03]. We added this result in results and discussion section.

Now it reads

In Results;

Moreover, the increase in sCr between preoperative and discharge time point was greater in control [0.35 (0.26‒0.47)] than RIPC group [0.30 (0.22‒0.42)] (p=0.03).

In Discussion;

This study demonstrated that RIPC attenuated the decrease in remnant kidney function in donors after donor nephrectomy. The postoperative sCr level after donor nephrectomy was statistically lower in the RIPC group, and the low sCr was maintained until 1 year after operation. The increase in sCr level between preoperative and at discharge was significantly greater in control group. Moreover, the proportion of kidney donors with sCr concentration >1.4 mg/dL at discharge was lower in RIPC group. Donors with serum creatinine within normal range at discharge had lower prevalence of chronic kidney disease (n = 8, 5.4%) than donors with sCr concentration >1.4 mg/dL (n = 6, 26.1%) (P = 0.003) after 1 year. Moreover, the recovery of eGFR was more rapid in the RIPC group than in the control group among donors during postoperative 1 month

Furthermore there are some other points that need clarification or improvement…

Introduction

Point 2.  Last paragraph of introduction.

 Studies on the effects of RIPC on post-transplant function in renal transplant recipients have yielded conflicting results [23,24]. Moreover, no well-designed clinical trial to date has assessed the effects of RIPC on remaining kidney function in KT donors. The anti-inflammatory effects inflicted byof RIPC [25] suggest that it may prevent additional  injury in the remainingnant donor kidney. In addition, ischemic preconditioning of donor  organs was found to reduce ischemic reperfusion injury and improve outcomes after liver transplantation [26,27]. However, it is unclear whether RIPC can reduce ischemic reperfusion injury  in the renal graft. Therefore, this studywe hypothesized that RIPC may attenuates the decrease in remnant  kidney function in donors following living donor nephrectomy. Additionally, this studywe assessed  the effects of RIPC on post-transplant renal function of recipients following living donor KT

Response 2. Thank you for kind suggestion. We revised that paragraph as your suggestion.

Methods section

Point 3. Please follow CONSORT guidelines. State clear inclusion and exclusion criteria.

Response 3. We apologize unclear description of inclusion and exclusion criteria. We enrolled a living donor whose age was between 20 and 60 years old. We excluded the donors with a history of acute kidney injury, morbid obesity, liver cirrhosis, hypertension, diabetes mellitus, hypoalbuminemia (<3.0 g/dL), hypoproteinemia (<6.0 g/dL), anemia (hemoglobin <10.6 g/dL, hematocrit <30%), or low body weight (body mass index <18 kg/m2). Recipients of enrolled donors were assigned to the group which their donors were allocated into. But two recipients were aged below 8 years, so we excluded those two recipients from final analysis of recipients.

Now it reads as below,

All living kidney donors aged 20–60 years for elective KT were included. Donors with a history of acute kidney injury, morbid obesity, liver cirrhosis, hypertension, diabetes mellitus, hypoalbuminemia (<3.0 g/dL), hypoproteinemia (<6.0 g/dL), anemia (hemoglobin <10.6 g/dL, hematocrit <30%), or low body weight (body mass index <18 kg/m2) were excluded. The recipients were included according to the matched donors. Recipients were divided into control or RIPC group according to their donor group. Two recipients in the control group were excluded from the analysis of outcomes because they were aged<8 years.

Point 4. It is unclear to me what the authors mean by: One of the tables was delivered to an anesthesiologist who was….

Response 4. We apologize for the confusion. Authors decided to remove this sentence. Now the randomization section was summarized as below;

A total of 170 (106 related and 64 unrelated) donors were randomized 1:1 to the RIPC and control groups using a randomization table generated using R (version 2.13.1) by a trial administrator who was independent of all other aspects of the trial. Surgeon, donors, and investigators were blinded to randomization.

Point 5. In the control group, the cuff… but the pressure was not increased. Please replace by: the cuff was not inflated.

Response 5. Thank you for kind suggestion. We changed the sentence as your suggestion.

Point 6. Donor kidney was preserved using the simple cold storage technique with histidine-tryptophan-ketoglutarate solution. Please remove the simple and technique. What company manufactures the HTK used?

Response 6. Thank you for correction. The name of the HTK solution is “Custodiol”, made by Essential Pharmaceuticals.

Point 7. Cystatin C was measured only in donors at the end of operation. Remove only and replace by: Additionally serum cystatin C was measured at the end of surgery

Response 7.  Thank you for kind suggestion. We changed the sentence as your suggestion.

Point 8. What is the baseline sample time point both in donors and recipients? The authors have written it in their response but still cannot find it in the manuscript. Were dialysis patients dialysed before or after this baseline.

Response 8. Mostly, baseline serum creatinine and eGFR was measured the day before the operation.

For donors,

Renal function of donors was analyzed by measuring serum creatinine (sCr) concentration and estimated glomerular filtration rate (eGFR), based on sCr [28] at 1, 3, and 7 days; and 1, 6, and 12 months after donor nephrectomy. Additionally, serum cystatin C was measured at the end of surgery. The baseline eGFR in donors was measured the day before the operation.

For recipients

The graft function of recipients was assessed after surgery by measuring sCr concentration and eGFR based on the Chronic Kidney Disease Epidemiology Collaboration (CKD-EPI) equation at 1, 3, and 7 days; and 1, 6, and 12 months after transplantation. The baseline eGFR was measured the day before the operation after dialysis. 

Point 9. please mention immunosuppressive protocol in short.

Response 9. Thank you for kin suggestion.

We added this paragraph in the section of 2.2 Recipients at methods section.

The main immunosuppressive protocol consisted of basiliximab as an induction agent, and maintenance immunosuppressants consisted of a combination of a calcineurin inhibitor (tacrolimus or cyclosporine), mycophenolic acid, and prednisolone. As another option, recipients (n=21) with immunologic risk factors (highly sensitized patients or re-transplant recipients) and those with complications due to long-term use of steroids received rabit anti-thyomocyte globulin (Thymoglobulin®, Genzyme, Cambridge, MA, USA) as an induction regimen, and maintenance immunosuppression included tacrolimus, mycophenolic acid, and early steroid withdrawal in a week.

Point 10. The 170 recipients were included according to the matched donors. Recipients were divided  into control or RIPC group according to their donor group, and retrospectively evaluated by review  of medical records. Two recipients in the control group were excluded from the analysis of outcomes  because they were aged<8 years. The donor kidney was inserted into the iliac fossa of the recipient. The iliac artery was mobilized, and about 3 cm was prepared for anastomosis. The external iliac vein  was also mobilized for end-to-side anastomosis by dividing the lymphatics overlying this vein, with  or without dividing the internal iliac vein at its origin. The donor renal vein was anastomosed to the  external iliac vein, and the donor artery was anastomosed to the recipient common or external iliac  artery in an end-to-side manner. The ureter was spatulated, and an end-to-side anastomosis was  completed to the bladder mucosa.

To my opinion the red part can be replaced by: kidney transplantation was performed by local standardized protocol. What do the authors mean with spatulated?? Don’t they mean cannulated?

Response 10. Thank you for kind suggestion. We changed the description of process of kidney transplantation as you suggested.

Point 11. For donors, Hartman solution or plasma solution was primarily administered, concomitantly with 5% albumin and mannitol. For recipients, plasma solution with 20% albumin and mannitol were used. For donors, hypotension was defined as mean blood pressure decreased below 65 mmHg and hypertension as blood pressure exceeding > 20 mmHg of baseline. For recipients, blood pressure change of more than 20% from baseline level was diagnosed as hypotension and hypertension. Both donors and recipients were treated with inotropic or vasopressors bolus administration for hypotension, and nicardipine or labetalol was used to treat hypertension. Transfusion was performed when hemoglobin level was < 8 g/dL.

Hartman solution replace by Ringers Lactate

plasma solution, what do you mean with this? Plasmalyte?

Response 11. Thank you for kind suggestion. Plasma solution (CJ Plasma solution A inj, CJ HealthCare®, Republic of Korea) is a kind of balanced crystalloid, which has same composition with plasmalyte. We corrected the sentence as you suggested.

Point 12. Suggestion to replace the red sentence If hypotension occurred, both donors and recipients were treated with inotropics or vasopressors… (Which ones?????). In case of hypertension nicardipine or labetolol was used.

Response 12. If hypotension occurred, both donors and recipients were treated with inotropics and/or vasopressors, as appropriated.

Point 13. Was the first step in case of hypo/hypertension not to adjust the depth of anesthesia/analgesia if possible??

Response 13. In our institution protocol, we use inotropics and/or vasopressors.

Point 14.  Primary outcome was post-nephrectomy and post-transplantation renal function according to RIPC in donors and recipients. Renal function in donors and recipients was assessed with the sCr concentrations and eGFR at discharge day as primary outcome measurements. Secondary outcome measurement in donors was the proportion with chronic kidney disease, defined according to Kidney Disease Improving Global Outcomes (KDIGO) criteria, 1 year after surgery. For recipients, time required for a 50% decrease in baseline sCr concentration; eGFR 12 months after KT; the proportion with delayed graft function; and the proportions with acute rejection and graft failure during the first 12 months were investigated as secondary outcome measurements. Delayed graft function was defined as a<10%/day reduction in sCr concentration on 3 consecutive days during the first week after KT. Rejection was defined as any biopsy-proven or biochemical rejection treated with pulsed methylprednisolone. eGFR at discharge and 1 year postoperative was adjusted by donor age and donor eGFR.

 Please replace by:

Primary outcome measure was the remaining kidney function post-nephrectomy in living kidney donors reflected by sCr levels and eGFR at day of discharge. Secondary outcome  measurements were: in donors, incidence of chronic kidney disease 1 year after surgery, according to the Kidney Disease Improving Global Outcomes (KDIGO) criteria and in recipients time required to a 50% decrease in baseline sCr concentration; eGFR 12 months after KT; incidence of delayed graft function, acute rejection and graft failure the first 12 months post transplantation. Delayed graft function was defined as a failure to<10%/day reduction in sCr concentration on 3 consecutive days during the first week after KT. Rejection was defined as any biopsy-proven or biochemical rejection treated with pulsed methylprednisolone. eGFR at discharge and 1 year postoperative was adjusted by donor age and donor eGFR.

Response 14. Thank you for kind suggestion. We corrected the paragraph as you suggested.

Point 15.  Delayed graft function was defined as a<10%/day reduction in sCr concentration on 3 consecutive days during the first week after KT. How did you handle the patients with excellent function on day 2 which did not show further reductions on day 3…..

Response 15. We classified those patients into a delayed graft function. Overall, reduction in serum creatinine less than 30% of baseline until postoperative day 3 was classified into delayed graft function.

Point 16.  Rejection was defined as any biopsy-proven or biochemical rejection. What do you mean with biochemical rejection.Normally in KT trials acute rejection is defined as biopsy proven and treated

Response 16. We apologize incomplete description on acute rejection. Authors corrected the sentence as below;

Rejection was defined as biopsy-proven rejection treated with pulsed methylprednisolone. Biopsy-proven rejection was defined as any rejection grade according to the Banff criteria, based on histopathological appearance of a needle core biopsy of the transplant kidney.

Point 17.  eGFR at discharge and 1 year postoperative was adjusted by donor age and donor eGFR. What do you mean with this… eGFR in recipients? If so how was this performed???

Response 17. We apologize our incomplete explanation on the statistical methods. Authors used a linear regression model to predict eGFR at discharge and postoperative 1 year in recipients and adjust them using warm ischemic time, preoperative sCr, age and eGFR of donor. Donor age and eGFR were known as potential predictors of recipient’s eGFR after transplantation (NDT Plus 2010;3(Suppl. 2): ii2–8, Transpl Int 2007;20:62–72, Nephrol Dial Transplan 2007;22:3646–51). Warm ischemic time and preoperative sCr were different between two groups. Authors corrected the Table 3.

Table 3. Outcomes in kidney transplant donors and recipients.

Donors

Control group (n=85)

RIPC group (n=85)

P-value

Progression to chronic   kidney disease, n (%)

6 (7.1)

8 (9.4)

0.78

Proportion with sCr ≥1.4   mg/dL at discharge, n (%)

17 (20.0)

6 (7.1)

<0.05< span="">

Recipients

Control   group (n=83)

Treated   group (n=85)

P-value

Time to 50% decrease in   sCr, days

1.1 ± 0.6

1.4 ± 3.2

0.58

Delayed graft function, n   (%)

0 (0.0)

2 (2.4)

0.49

Acute rejection, n (%)

5 (6.0)

5 (5.9)

1.00

Graft failure, n (%)

1 (1.2)

0 (0.0)

0.99

eGFR at discharge,   mL/min/1.73 m2

87.9 ± 17.7

83.9 ± 19.4

0.18

 Mean difference1

-3.23 (-8.62‒2.16)

0.14

eGFR at postoperative 1   year, mL/min/1.73 m2

77.0 ± 24.5

79.5 ± 18.2

0.49

 Mean difference1

3.45 (-3.1110.01)

0.29

RIPC, remote ischemic preconditioning; sCr, serum creatinine; eGFR, estimated glomerular filtration rate. eGFR was calculated using the CKD-EPI equation. 1Difference in means (95% CI) adjusted for warm ischemic time and preoperative serum creatinine, age and baseline eGFR of donor.

Point 18. time required to a 50% decrease in baseline sCr concentration…

When was baseline sCr measured? Was this the same timepoint for all patients? Were dialysis patients dialyzed before or after.. This significantly influences the outcome parameter…

 Response 18. The baseline eGFR was measured the day before the operation after dialysis. 

Point 19. The authors state: We identified the time required for a 50% decrease in baseline sCr estimated by graphical inspection. This is rather vague to me please clarify.

Response 19. We plotted the measured sCr according to the time points and inspected the time point when sCr was decreased by 50% visually for each patient. After that, we averaged the time points when sCr decreased below 50% of baseline. 

Point 20. Regarding the sample size calculation: Sample size was determined as described previously [29]. Briefly, that study showed that the average eGFR calculated by the CKD-EPI equation on postoperative day 7 was 55 ± 11 mL/min/1.73 m2.

I cannot find these data in the article you are referring to…..

Response 20. We apologize our incomplete description on sample size. The sample size was determined based on our previous study [Bang JY et al, Medicine (Baltimore) 2017 Feb; 96 (5): e6037]. We obtained the minimum eGFR during the post-nephrectomy 1 weeks from the 1647 donors of kidney transplantation through the preceding study. But that information was not shown in the article [Bang JY et al, Medicine (Baltimore) 2017 Feb; 96 (5): e6037]. The mean of minimal eGFR (standard deviation) was 55 (11) mL/min/1.73m2. Based on this result, we calculated the sample size to test the hypothesis that RIPC could increase the eGFR by 10% (alpha=0.05, beta=0.8) The estimated number was 75.9. Considering the drop-out rate of 10%, we determined the 85 patients in each group.

Authors revised the description as below;

Sample size was determined based on the results of preceding study, which the data were not shown [29]. Briefly, we obtained the minimum eGFR (calculated by the CKD-EPI equation) during the post-nephrectomy 1 weeks from 1647 donors of kidney transplantation through the preceding study.  The mean ± SD of minimal eGFR during the post-nephrectomy 1 week was 55 ± 11 mL/min/1.73 m2.

Point 21. Please describe how the Multivariate analysis was performed

Response 21. We apologize our incomplete description. Authors additionally described how the multivariate analysis was performed in the statistical analysis section. Now it reads,

Multivariate analysis was conducted for each variable with p < 0.1 in the univariate analysis.

Point 22. Flowchart of the trial……This is called a consort diagram.

Response 22. We are so sorry for that mistake. We corrected “Flowchart of the trial” with “Consort diagram” as you suggested. 

Point 23. Table 1

Operation site replace by: kidney

Plasma solution. Do you mean balanced cristalloids?

Levophed. Please replace by noradrenaline/norepinephrine

Response 23. We are so sorry for that mistake. We corrected those wrong words as you suggested. 

Plasma solution (CJ Plasma solution A inj, CJ HealthCare®, Republic of Korea) is a kind of balanced crystalloid, which has same composition with plasmalyte.
